# Role of Insect and Mammal Glutathione Transferases in Chemoperception

**DOI:** 10.3390/biom13020322

**Published:** 2023-02-08

**Authors:** Mathieu Schwartz, Valentin Boichot, Stéphane Fraichard, Mariam Muradova, Patrick Senet, Adrien Nicolai, Frederic Lirussi, Mathilde Bas, Francis Canon, Jean-Marie Heydel, Fabrice Neiers

**Affiliations:** 1Laboratory: Flavour Perception: Molecular Mechanims (Flavours), INRAE, CNRS, Institut Agro, Université de Bourgogne Franche-Comté, 21000 Dijon, France; 2Laboratoire Interdisciplinaire Carnot de Bourgogne, UMR 6303 CNRS, Université de Bourgogne Franche-Comté, 21078 Dijon, France; 3UMR 1231, Lipides Nutrition Cancer, INSERM, 21000 Dijon, France; 4UFR des Sciences de Santé, Université de Bourgogne Franche-Comté, 25000 Besançon, France; 5Plateforme PACE, Laboratoire de Pharmacologie-Toxicologie, Centre Hospitalo-Universitaire Besançon, 25000 Besançon, France

**Keywords:** insects, glutathione transferase, olfaction, taste, chemosensory organs, detoxification, evolution, flavor

## Abstract

Glutathione transferases (GSTs) are ubiquitous key enzymes with different activities as transferases or isomerases. As key detoxifying enzymes, GSTs are expressed in the chemosensory organs. They fulfill an essential protective role because the chemosensory organs are located in the main entry paths of exogenous compounds within the body. In addition to this protective function, they modulate the perception process by metabolizing exogenous molecules, including tastants and odorants. Chemosensory detection involves the interaction of chemosensory molecules with receptors. GST contributes to signal termination by metabolizing these molecules. By reducing the concentration of chemosensory molecules before receptor binding, GST modulates receptor activation and, therefore, the perception of these molecules. The balance of chemoperception by GSTs has been shown in insects as well as in mammals, although their chemosensory systems are not evolutionarily connected. This review will provide knowledge supporting the involvement of GSTs in chemoperception, describing their localization in these systems as well as their enzymatic capacity toward odorants, sapid molecules, and pheromones in insects and mammals. Their different roles in chemosensory organs will be discussed in light of the evolutionary advantage of the coupling of the detoxification system and chemosensory system through GSTs.

## 1. Introduction

Evolution shapes all living organisms to perceive chemicals, enabling them to detect nutritive compounds and avoid toxic compounds. Although chemical detection supports several biological functions related to chemical communication in animals, the primary function of this ancient sense was most likely to find nutritive molecules. According to the Red Queen hypothesis [1], it can be hypothesized that the appearance of detoxification systems results from developments in defensive systems, such as the synthesis of toxic compounds. In response to this new pressure of selection, predators have evolved and developed defensive mechanisms capable of taking in toxic xenobiotics. In this context, the extension of the organism’s detoxifying capacity should have also enabled species to extend their food sources due to their new capacity to metabolize xenobiotic compounds within these food sources. Xenobiotics are molecules that are not used as building blocks for biological macromolecules and also do not provide energy. Glutathione transferases (GSTs; EC 2.5.1.18), originally discovered as detoxification enzymes [2,3], are one of the key enzymes involved in the metabolization of endogenous and exogenous molecules. GSTs are found in most living organisms, including insects and mammals. Xenobiotic molecules include chemicals naturally present in food sources as well as molecules produced industrially as pollutants and pesticides. Most flavor molecules can also be considered xenobiotics, regulated by GSTs and other detoxifying enzymes through the xenobiotic detoxifying network [4,5], such as glutathione transferases (GSTs; EC 2.5.1.18). GSTs, originally discovered as detoxication enzymes [2,3], are found in most living organisms, including insects and mammals. The best known function of GST is to catalyze the conjugation of reduced glutathione (GSH) to xenobiotic electrophilic centers, resulting in an increase in the hydrophilicity of xenobiotic compounds and thereby facilitating their elimination from the body. This function is shared by two entirely distinct superfamilies of enzymes [6], one microsomal (membrane-associated eicosanoid and glutathione metabolism, also called MAPEG), and the other soluble, also called canonical. These two superfamilies are not evolutionarily related and are found both in mammals and insects; moreover, in general, the number of microsomal GSTs is much lower than that of canonical GSTs. A third type of GST can be considered: mitochondrial Kappa GST [7]. Both kappa and canonical GSTs present a thioredoxin-like domain recognizing the GSH motif but are differently organized in the overall fold, suggesting their parallel evolution from a thiol-disulfide oxidoreductase progenitor [8]. Kappa GSTs are found in prokaryotes and eukaryotes, including mammals, but not in insects. However, Kappa GSTs have been identified in other arthropods and crustaceans [7,9]. For example, Acari, *Ixodes scapularis*, presents two Kappa GSTs involved in glutathione conjugation, and canonical GSTs also catalyze peroxide reduction [10], dehydrochlorination [11], and isomerization [12,13]. Moreover, they may trap substrates in the absence of enzymatic activity, which enables the sequestration of toxic compounds; this function is called ligandin [14]. Canonical GSTs are subdivided into classes, designated by the names of the Greek letters—Alpha, Delta, Epsilon, etc.—abbreviated as Roman capitals A, D, E, etc. Class members are distinguished by Arabic numerals [15]. GST members exhibit their own distinct tissue-specific expression patterns, suggesting that they have different functions. This differential pattern of expression is observed both in mammals [16] and insects [17]. It is not surprising that GSTs are preferentially expressed in insect detoxification organs, such as the fat body, midgut, or epidermis [18], and in the mammalian liver [19]. Chemosensory organs are exposed to the external environment and, thus, to xenobiotics. Consequently, GST expression is also observed in these organs, which represent specific places in the animal body. The detoxifying system can advantageously protect the chemosensory system, and additionally or as a result of its activity, it modulates chemosensory detection. The molecular organization of the chemosensory organs has been well described in both mammals and insects (the main laboratory models are rodents and *Drosophila*). In this review, we detail the knowledge supporting the involvement of GSTs in chemoperception, describing their localization in these systems as well as their enzymatic capacity toward odorants, sapid molecules, and pheromones in insects and mammals.

## 2. The Involvement of GSTs in Mammalian Chemoperception

### 2.1. Chemoperception in Mammals

Chemoperception in mammals involves three components: olfaction, taste, and trigeminal sensations [20,21]. Olfaction is a fundamental sense that enables animals to locate their food and their sexual partners and to warn of danger, which is also of great importance for their well-being through the hedonic tone of food. In mammals, olfaction is assured by the olfactory system located in the nasal cavity. Odorant compounds are small, volatile chemicals, generally of a hydrophobic nature, that enter the body through the nostrils and solubilize within the nasal mucus. Odorants bind to olfactory receptors (ORs) located on the olfactory receptor neurons (ORNs) present in the main olfactory epithelium. Indeed, in addition to the olfactory epithelium, mammals have several accessory olfactory organs, such as the vomeronasal organ, the septal organ of Masera, and the Grueneberg ganglion, which all contain ORNs. These organs exhibit overlapping functions with the main olfactory system, but their specificities are not fully characterized. Each olfactory receptor neuron expresses only one type of OR [22]. ORNs are connected to the olfactory bulb, where the olfactory signal is processed and further transmitted to higher brain regions. This organization transforms a chemical signal into an electrical signal thanks to an efficient combinatorial code of odors permitted by the high diversity of olfactory receptors (380 for humans) [23].

In contrast to olfaction, which is important for various functions such as food search and enjoyment, reproduction, and survival, the sense of taste in mammals is exclusively devoted to the evaluation of food quality and is closely related to feeding behavior [24]. Taste compounds are divided into five qualities, namely, sweet (sugars), bitter (various compounds of organic or inorganic nature), sour (acidic compounds), salty (ionic inorganic compounds such as Na^+^), and umami (amino acids). In mammals, taste sensations result from the activation of taste receptors by taste compounds. Taste receptor cells assemble at the surface of the tongue or palate into small structures called “taste buds”, composed of approximately 100 cells. These taste buds are found in epithelial structures called “papillae”, classified into different types and present different structures and locations at the tongue surface. Papillae include fungiform, circumvallate, and foliate papillae. Taste receptor cells are linked by afferent nerves to the geniculate and petrosal ganglions, which mediate taste signals to the brain stem. Taste receptors from the T1R and T2R families are part of the class C and class A GPCR families, respectively [25]. They assemble through different functional homo- or heterodimers for T1Rs that are able to detect sweet (T1R2 + T1R3) and umami (T1R1 + T1R3) compounds. Monomeric T2Rs enable the detection of bitter molecules. Two other families of receptors have been proposed for the detection of sourness (PKD2L1) and saltiness (ENaC) [24].

In addition to the five qualities of taste, a few studies have evidenced a sixth sensation, which is more of a modulation of some of the basic tastes, namely the kokumi taste [26]. Kokumi means “rich taste” in Japanese and is associated with the reinforcement of umami, sweetness, and saltiness due to the presence of kokumi compounds. Kokumi compounds include both nonpeptide compounds and peptide compounds such as gamma-glutamyl peptides such as glutathione [27]. Kokumi compounds have been proposed to activate the calcium-sensing receptor (CaSR) to elicit the kokumi sensation [28]. Kokumi is an increasingly studied topic of research with great potential for flavor enhancement and food product development [26].

In addition to olfaction and taste, a third chemosensory sensation is related to the trigeminal system and enables the detection of trigeminal compounds such as astringent plant polyphenols, compounds present in foods eliciting a sensation of cooling (menthol) or burning (capsaicin), or carbonated drinks containing CO_2_ that trigger a prickly sensation [21]. Trigeminal compounds mainly stimulate transient receptor potential (TRP) channels present in sensory neurons of the oral mucosa, which convert chemical signals into electric activity. The identification of astringent compounds may either involve mechanoreceptors detecting changes in the friction forces at the surface of the oral mucosae or the transmembrane mucin MUC1, as recently proposed by our group [29].

Olfaction, taste, and trigeminal sensations can be modulated by the metabolic activities present in the vicinity of the chemosensory receptors [30]. This metabolism involves enzymes such as glutathione transferases (GSTs). These have shown to be linked to both olfaction and taste perception, as explained in the following section.

### 2.2. Roles of GSTs in Mammalian Chemoperception

Many drug-metabolizing enzymes are present in the nasal cavity, within the nasal epithelium and mucus [31,32], as well as in the oral cavity, within the saliva and the oral mucosa [33,34]. These enzymes assure the protection of the epithelia and, notably, the olfactory receptor neurons from damage provoked by xenobiotics entering the nasal (or oral) cavity. In addition to their role in detoxification, some of these enzymes handle flavor molecules, thus producing flavor metabolites in the oronasal sphere [35,36,37,38]. All these enzymes, in addition to proteins able to bind flavor compounds such as odorant-binding proteins (OBPs), have an impact on both the quality and quantity of flavor compounds that activate chemoreceptors [30]. These molecular mechanisms have been named “perireceptor events” because they occur in the environment surrounding the chemoreceptors. In the context of mammalian olfaction, a body of evidence indicates that glutathione transferase is an important perireceptor enzyme acting on odorant detection modulation. Additionally, pioneering studies have started to investigate the potential role of glutathione transferases in taste perception. Both of these aspects are treated in this part of the review. Among the seven canonical mammalian GSTs identified thus far (alpha, mu, theta, pi, omega, sigma, and zeta [15]), alpha, mu, and pi classes of GSTs are common [39] among mammals [40].

In 1992, Ben-Arie and coworkers showed that the olfactory epithelium is the extrahepatic tissue in a rat model that exhibits the highest glutathione transfer activity with the chemical substrate chlorodinitrobenzene (CDNB), thus showing the strong expression of GSTs and suggesting a potential role in olfaction [41]. All GST classes were shown to be expressed in chemosensory organs in different mammalian species (Table 1). Glutathione transferase of the mu class was demonstrated in the rat nasal mucus and olfactory epithelium [4], supporting a previous study already showing GST expression in rat chemosensory mucosae [42]. Additionally, in the same study, the ability of recombinant rat GSTM2 to catalyze the transfer of glutathione to various odorant compounds, including aldehydes, ketones, and epoxides, was shown. Using a competition assay, GSTM2 was also found to be able to bind a large variety of odorants, most likely due to its ligandin properties [4]. Similar results were obtained for human GSTA1 and GSTP1, which are two GSTs that play roles in glutathione and ligand transfer for many odorous molecules and are expressed in the human respiratory epithelium [43]. Immunohistochemistry experiments showed the localization of human GSTs in ciliated cells (at the surface of the epithelium from the human olfactory vicinity), thus facilitating the entry of odorants into the epithelium. Furthermore, the first X-ray structure of a GST bound to a metabolized odorant enabled a fine analysis of its active site and its capacity to specifically recognize odorant molecules [43]. The crystal structure of human GSTA1 bound to the metabolite glutathionyl-dihydrocinnamaldehyde was analyzed and revealed the ability of the hydrophobic site of GSTA1 to strongly adapt to small hydrophobic volatile compounds. This property facilitates the binding of cinnamaldehyde and promotes the formation of the glutathione conjugate through a nucleophilic substitution, suppressing the carbon double bond of cinnamaldehyde [43].

GST expression levels were also shown to be associated with olfactory dysfunction. Zinc deficiency is linked to olfactory and gustatory dysfunction in mammals [44,45]. Interestingly, in rats, zinc deficiency is associated with a reduction in GST mRNA in some cell types of the olfactory epithelium (supporting cells [46]).

The involvement of GSTs in odorant metabolism has become increasingly documented; however, less evidence is available concerning the significance of these molecular mechanisms in odorant perception. In this context, experiments with a rabbit model enabled us to reveal some clues on the role of GSTs in odorant signal termination. It was shown that the mammary pheromone, corresponding to the compound 2-methylbut-2-enal (2MB2), triggers the grasping of the mother rabbit mammae. Newborn rabbits are blind, and suckling-related behavior allows them to survive despite the short time that allows the mother to feed them [47]. This chemically triggered behavior is critical for pups, which are constrained to finding nipples within the five minutes of daily nursing. It has been shown that the mammary pheromone is metabolized to a glutathione conjugate in the olfactory epithelium of newborn rabbits, in accordance with a high early expression of glutathione transferases in this tissue [48]. Furthermore, it has been shown that this metabolism is also present in the nasal mucus of newborn rabbits due to the presence of glutathione transferases in this biological fluid based on proteomic analysis [49]. Additionally, the deregulation of this metabolism by in vivo washing of the nasal mucus, thus diminishing the glutathione conjugation of 2MB2, led to increased sensitivity of the behavior response in newborns exposed to the mammary pheromone. Similar results were obtained when the 2MB2 metabolism was reduced due to competition with another odorant substrate catalyzed by the same enzyme [50]. Decreasing the glutathione conjugation of the mammary pheromone allowed us to record behavioral responses with concentrations of mammary pheromone that were usually inactive. Glutathione conjugation to the mammary pheromone modifies its structure and thus terminates the odorant signal. This metabolization thus enables the newborn rabbit to remain responsive to the mammary pheromone by reinitializing the chemical signal. Rabbit GSTs catalyze glutathione conjugation with 2MB2 but can ensure the role of odorant signal termination for a wide range of other odorant compounds in mammals, including humans.

In addition to the nasal cavity, GSTs are expressed in the oral cavity, particularly in taste bud cells [51]. In rats, GSTM and GSTP were found to be expressed in both circumvallate and foliate papillae. The expression seems to be GST member-dependent; GSTA is not found in these papillae. Moreover, the results obtained in human and rat olfactory epithelia show differences with regard to the type of GST expressed in this tissue depending on the species, highlighting differences in their expression in chemosensory tissue between mammalian species [4,43]. GST was also found to be expressed in mammalian saliva, including human saliva [52]. Glutathione is found at a concentration of approximately 1 g/L in human saliva [53], enabling the GSH saturation of GSTP, the main human salivary GST, and then allowing it to catalyze glutathione conjugation at the maximal rate. Additionally, this salivary expression was shown to be modulated in humans in association with food behavior. Indeed, increased GST expression in response to specific diets, such as those rich in broccoli or coffee, was observed [52,54]. Recently, a study explored a possible link between GST expression in saliva and bitter taste perception [55]. GSTA1 and GSTP1 were identified in a cohort of 104 people, all of whom expressed the 2 GSTs in saliva. Additionally, people exhibiting ageusia or dysgeusia, included in this study, showed significantly lower salivary GSTA1 levels than those in the saliva of the control group, suggesting possible relationships between salivary GST levels and taste function. In the same study, GSTA1 and GSTP1 interacted with various bitter compounds, including flavonoids and isothiocyanates. This last family of compounds is metabolized within the saliva [55]. The X-ray structures obtained between GSTs and isothiocyanates showed that different binding sites exist, whether the interactions imply glutathione conjugation (binding in the active site) or covalent adduction to an exposed cysteine in the GSTA1 ligand site.

All these elements suggest the involvement of GSTs in mammalian taste perception, in addition to olfaction. In addition, the level of expression of GST is correlated with disorders related to both taste and smell.

**Table 1 biomolecules-13-00322-t001:** Location and classes of GSTs within the mammal’s chemosensory organs.

Mammal Species	Location	GST Classes	Ref.
*Canis lupus familiaris*	Saliva	Alpha, Mu, and Omega	[56]
*Equus caballus*	Saliva	Pi	[57]
*Homo sapiens*	Olfactory mucus	Alpha and Pi	[58,59]
	Olfactory epithelium	Alpha, Mu, and Pi	[43]
	Saliva	Alpha, Kappa, Mu Omega, Theta, and Pi	[60,61]
*Mus musculus*	Sensory cilia	Alpha, Kappa, Mu, Omega, Tau, and Zeta	[62]
	Saliva	Omega	[63]
*Oryctolagus cuniculus*	Olfactory mucus	Alpha, Mu, and Pi	[49]
*Ovis aries*	Saliva	Alpha	[57]
*Rattus norvegicus*	Sensory cilia	Alpha and Mu	[64]
	Olfactory epithelium	Alpha, Mu, and Pi	[4,41]
	Olfactory mucus	Alpha, Mu, and Pi	[4]

## 3. The Involvement of GSTs in Insect Chemoperception

### 3.1. Chemoperception in Insects

Insects constitute the largest class of living animal species. Due to their small size, they have developed multiple mechanisms to limit toxic xenobiotic effects, including the enhancement of metabolic detoxification [65,66], which reduces penetration through the cuticle, or behavioral avoidance [67]. Insects can taste through many parts of their body. The proboscis organs used for feeding and sucking allow insects to taste food during ingestion before it reaches the digestive system. In addition to the proboscis, insects are able to detect tastants with their legs, wings, and ovipositor organs. Consequently, due to their small size compared with food, they are generally already in contact with it before ingesting it, making it advantageous to taste it with their legs before eating or with the ovipositor organ before laying eggs. Although the taste systems of mammals and insects evolved independently, they enable the detection of similar qualities, including sweet, salty, and bitter stimuli. Insects are able to detect carbonation as a taste modality through gustatory neurons [68], similar to mammals, through carbonic anhydrase IV, which produces protons that activate a proton-gated channel [69]. Interestingly, carbonation detection is also possible in both mammals and insects through the olfactory system [69]. Insects have gustatory sensory neurons that mediate the recognition of water [70]; to date, it has not been established whether other animals can taste the water. In mammals, taste receptors are not hosted by neurons. Neurons are in contact with the taste cells that carry gustatory receptors and are located in taste buds. In insects, gustatory receptors are directly carried by gustatory neurons, in contrast to vertebrate gustatory neurons, which are housed in cells that are indirectly in contact with neurons. Gustatory neurons are housed within the hundreds of gustatory sensilla distributed on the surface of the different sensory organs except the proboscis, which also includes internal sensilla [71].

In insects, the equivalent of the mammalian nose is the antenna and the maxillary palps. Although they do not exhibit any evolutionary relationship with mammals, olfaction is also supported by olfactory neurons in insects. Indeed, the olfactory sensilla cover the distal segment of the antenna, and the maxillary palps host the olfactory neurons. Dendrites of olfactory neurons that express olfactory neurons are located in the sensilla lymph within the sensilla. Odor molecules pass through pores or slits in the sensillum cuticle and enter the sensillum lymph [72]. Insect olfactory receptors are not homologs of vertebrate olfactory receptors [73], suggesting different evolutionary origins compared with those found in vertebrates. Consequently, although the organizational features of the olfactory systems of vertebrates and insects appear very similar, these structures may not share a common evolutionary heritage [74].

Insect GRs and ORs, which are membrane proteins, do not show any homology to those of vertebrates [75,76] and consequently do not belong to the GPCR family. However, both GRs and ORs evolved from an ancestral protein, and in addition to sharing the same sequence identity, they share the same inverted transmembrane topology as vertebrate olfactory GPCRs. The expansion of genes coding for the insect GR and OR has occurred only in insects [77]. In contrast to the monomeric ORs of vertebrates, insect ORs form heteromers with a conserved OR receptor also called Orco (i.e., the OR coreceptor). One specific OR is expressed in each insect olfactory neuron in addition to Orco, as in vertebrate olfactory neurons, where one specific OR is expressed in each olfactory neuron. It is unclear whether insect GRs can function alone as multimers or with other insect GRs due to the observation that multiple GR genes are expressed in a single GR neuron [78].

Even if the chemosensory systems in insects and mammals present similar biological organizations, they do not share any evolutionary links. However, in both chemosensory systems, glutathione transferases from the same common ancestor are expressed, most likely sharing similar physiological roles.

### 3.2. Roles of GSTs in Insect Chemoperception

In addition to the Omega, Sigma, Theta, and Zeta GST classes found in insects and shared with mammals, two other classes are observed: Delta and Epsilon GSTs. Delta GSTs are found in insects and are observed in a more general manner in arthropods, such as crustaceans [9]. Epsilon GSTs appear more specific to insects and were hypothesized to be insect-specific [79]. The numbers of Delta and Epsilon GSTs are variable from one insect species to another, mostly due to duplication events that occur in each insect species (Table 2). This gene-coding GST duplication might be associated with functional differentiation during insect evolution and is related to environmental adaptation. Gene duplication followed by sequence divergence is a key process during evolution, allowing the creation of novel gene functions [80]. Interactions of insects with plants, and especially plant chemicals and their adaptations to them, appear to be the most likely major driving force in herbivorous insect evolution [81]. Plant molecules can be toxic to insects, and consequently, GSTs and detoxifying enzymes are essential for insect survival. GSTs detoxify a broad range of plant molecules, generally with an overlap of GSTs for the same substrate [82,83,84]. Signatures of a positive selection of Delta GSTs suggest that they may have evolved under positive selection in the herbivorous [85] lineage after the transition of insects to herbivory > 350 Ma [86]. This adaptation phenomenon can be rapid; indeed, anthropological pressure toward insects for insecticide resistance has been suggested to promote *Musca domestica gst* gene amplification [87,88]. The main classes of GST diversification appear to be the Epsilon and Delta classes in various insects, such as *Anopheles gambiae*, *Drosophilia melanogaster*, or *Tribolium castaneum* [79,89]. In this context, it is not surprising to observe numerous insect adaptations toward insecticides [90,91] due to the Delta and Epsilon GSTs in the role of *A. gambiae* GSTE1 and GSTE2 in the DDT resistance [92]. The chemical resistance promoted by GSTs involves various chemicals, such as pyrethroids or neonicotinoids and 2,2-dichlorovinyl dimethylphosphate for *Diaphorina citri* [93] and *Rynochophorus phoenicis*, respectively [94]. In contrast, *Apis mellifera*, known to be highly sensitive to insecticides, presents only one Delta GST (including two isoforms) and no Epsilon GST. It is not excluded that some GSTs resulting from functional differentiation appear with different functions not related at all to xenobiotic metabolism, such as GSTE14 in *D. melanogaster*, which is involved in ecdysone biosynthesis [13,95,96]. Additionally, GSTs formed during the diversification process can also be specific in metabolizing some molecules without a functional overlap from other GSTs within the same insect species. For example, the deletion of epsilon and omega GSTs in the Asian gypsy moth, *Lymantria dispar*, affected its adaptability to salicin and rutin produced by its host, the poplar tree [97].

As shown in vertebrate chemosensory organs, GSTs are also expressed in insect chemosensory organs (Table 3). This expression is advantageous to protect these sensitive organs where neurons are directly exposed to xenobiotics. GSTs were shown to be expressed in the antennae of various orders of insect species, such as the dipteran *D. melanogaster* [17,134], various lepidopteran species [107,130,135,136,137,138,139] such as *Manduca sexta* [140] or *Spodoptera littoralis* [141], and in the Coleoptera antennae of *Agrilus planipennis* [142] or *Dendroctonus valens* [143]. Table 3 shows the diversity of insect species expressing GSTs within their sensory organs. Although a limited number of studies have analyzed GST expression, GSTs appear to be ubiquitously expressed in antennae. To support this hypothesis, GST expression in two particular insect species can be highlighted. The only Delta GST found in *A. melifera* is expressed in its antennae [144]. Ticks have a unique chemosensory organ presumed to function similarly to insect antennae, the fore-tarsal Haller’s organ. GSTs were found to be expressed in this organ of the dog tick, *Dermacentor variabilis* [145]. As in mammals, GSTs were proposed to protect the chemosensory organs so they could participate in odorant clearance and consequently signal termination. Antennal GSTs were shown to be active toward the model substrate CDNB (1-chloro-2,4-dinitrobenzene); for example, most antennal *Drosophila* GSTs [146,147]. The ability to conjugate CDNB was also observed for the antennal-specific GST identified in *Bombyx mori* [137]. The selective pressure to conserve efficient odorant clearance is crucial for flying insects, which need to reinitiate odorant perception as quickly as possible to follow the odorant volute. The detection of odorant food sources can be diversified, probably involving different GSTs. However, pheromone detection can also involve more specialized GSTs, as shown for an antenna-specific Delta GST found in *Manducta sexta*. This GST was shown to metabolize *trans*-2-hexenal, a plant-derived green leaf aldehyde known to stimulate the olfactory system of *M. sexta*. This GST was proposed to be involved in the signal termination of a complex mixture of aldehyde molecules forming the sex pheromone bouquet [140]. A delta GST found in the antennae of *Grapholita molesta* shows high activity toward a sex pheromone component, (*Z*)-8-dodecenyl alcohol [148]. The role of GST in sex pheromone detection is also supported by the differential expression of GST depending on the insect sex. For example, antennal-specific genes of a GST belonging to the Delta class were significantly more highly expressed in male *Helicoverpa armigera* antennae compared with females [139]. Despite all the different studies, to the best of the authors’ knowledge, the cellular localization of GSTs within the olfactory sensilla is not known to date. Moreover, expression within the sensory lymph, where the olfactory neurons are located, has not been validated experimentally. The same question about localization exists for insect gustatory sensilla. GSTs have been identified in diverse taste organs, such as the labellum, in insects belonging to the dipteran and ledidopteran orders (Table 3); however, the cell types and localization within the lymph of gustatory sensilla are not shown. Food containing bitter molecules such as glucosinolates and isothiocyanates led to GST overexpression in aphids in a general manner [149]. Additionally, other results have shown the same regulation in chemosensory organs. Isothiocyanates were shown to increase the expression of GST Delta in *Drosophila* labellum [5]. This modulation can be hypothesized to affect food habits. The results showed that the loss of bitter taste receptors observed in *D. suzukii* in comparison to *D. melanogaster* was proposed to contribute to the evolutionary shift in oviposition preference between the two species [150], knowing that ovipositor organs are taste-sensitive. In the same study, the taste difference between these two *Drosophila* species showed an associated differential expression of xenobiotic metabolism enzymes, including delta and epsilon GSTs. Again, this observation supports a direct link between these enzymes and the taste biochemistry in insects.

## 4. Discussion

GST appears to be a main actor in the mammalian and insect detoxifying systems. In insects, Delta and Epsilon GST diversification were shown to be associated with chemical resistance toward numerous molecules found naturally in plants, such as isothiocyanates, or resulting from human activity, such as pesticides. Interestingly, GSTs were found to be involved in the metabolism of similar molecules both in mammals and insects after the separation of these lineages [156]. Isothiocyanate molecules found in terrestrial plants were shown to modulate GST expression in mammals and insects [5,54,157]. These observations support the idea that GST gene diversification offers advantageous opportunities during evolution to build functional chemosensory systems to face molecular diversification in plant molecules. It may also have offered some species better adaptability to their environment, although this advantage is less important for species with a more specific ecological niche, such as *Apis melifera*. Interestingly, both olfactory and gustatory organs appeared independently in mammals and insects during their evolution and present an analogous organization as the chemosensory neurons wearing membrane olfactory receptors. In both cases, the receptors evolved from a different membrane protein ancestor, whereas the GSTs found in these two systems evolved from the same ancestral GST (homologous). In the insect lineage, GSTs of the Delta class have been found in almost all studies analyzing the contents of the antennae, indicating the importance of this class in the olfactory organs.

Crustaceans and insects share Delta and Epsilon GSTs, indicating their appearance before the separation of the lineages and consequently before the evolutionary elaboration of their olfactory systems, which did not share the same protein actors [158]. This observation supports that these specific GST classes did not appear to support the physiology of the chemosensory organs, and it is more likely that different members of existing classes of GSTs were randomly used in the chemosensory organs, highlighting the versatility of GSTs. One of the main biological functions of chemosensory GSTs is the protection of the chemosensory organs where neurons are exposed. The lifespan of olfactory neurons in rodents is approximately 40 days [159], although xenobiotic metabolism enzymes already contribute to extending this life; thus, neurons must continuously be replenished. In this context, it is not surprising to find numerous GST members as key players in xenobiotic metabolism in both mammalian and insect chemosensory systems (Table 3). In addition to chemoprotection, GSTs modulate their perception when these xenobiotic molecules are perceived by decreasing the xenobiotic concentration. This metabolic activity can also be involved in the termination of the signal if the metabolites are no longer perceived, or in the perception of a new signal if the metabolite is perceived by other chemoreceptors (Figure 1). To date, GSTs have only been shown to be involved in signal termination in mammals (rabbits) and insects. Due to the increased steric hindrance after glutathione conjugation, it is more likely that the chance of molecules binding receptors will decrease. However, it is not excluded that, after GST catalysis, those molecules can be perceived differently. Other GST activities, such as glutathione conjugation, are more susceptible to generating molecules that can be perceived, including isomerization activity. Indeed, chemosensory receptors are known to be stereospecific. Additionally, the ligandin capacity of GSTs has been proposed to help the diffusion of hydrophobic molecules as odorants in hydrophilic olfactory mucus surrounding the olfactory receptors [4]. A similar role in the sensillary lymph surrounding the insect olfactory receptors is not excluded if GSTs are expressed within the lymph. GST localization in sensilla lymphs is only supported by the observation of a signal peptide in some insect GSTs that are expressed in sensilla [141]. Interestingly, an increase in diffusion can also be proposed for all chemosensory molecules as a consequence of glutathione conjugation, as suggested for bitter molecules [55]. After glutathione conjugation, molecules are more hydrophilic. Thus, if it does not impact the ability of the molecules to bind to their receptors or their affinity ranges, they are most likely rapidly perceived.

To conclude, numerous questions about GSTs and their function in the chemosensory system remain open. For instance, their roles in trigeminal perception have never been studied to date in any organism. Numerous GST members are produced in bacteria, opening questions regarding their role in both mammalian and insect chemoperception in view of recent advances showing the potential roles of bacterial enzymes in human chemosensory perception [160,161] and insect chemoperception [162]. GST functions in chemosensory perception seem to be conserved in non-evolutionarily related chemosensory systems. Thus, the knowledge obtained from experimental data in one lineage should be tested in other lineages to enhance the understanding of their functions.

## Figures and Tables

**Figure 1 biomolecules-13-00322-f001:**
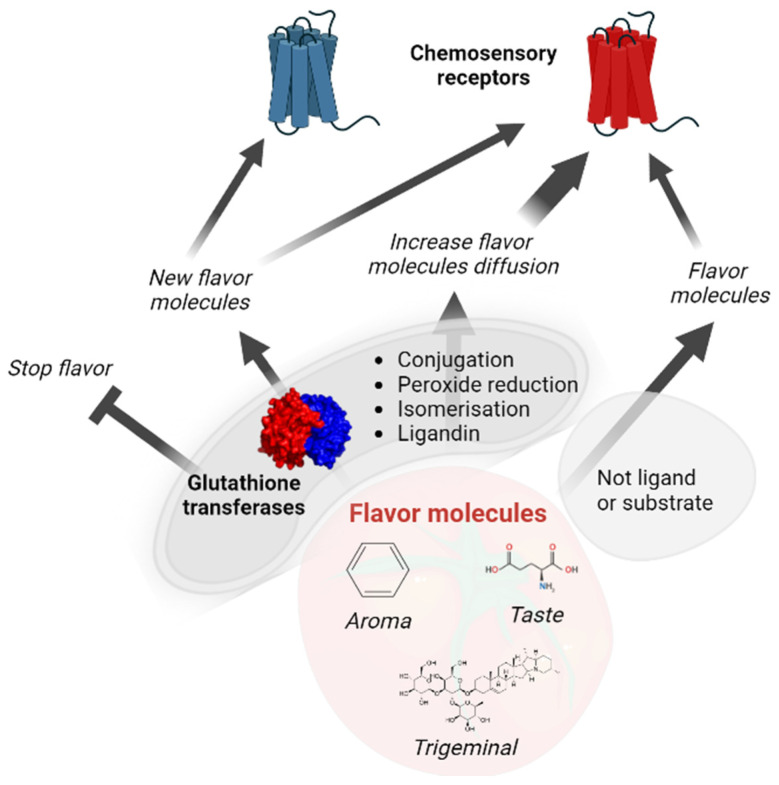
Role of glutathione transferases in chemosensory modulation.

**Table 2 biomolecules-13-00322-t002:** Number of identified canonical GSTs in different insect species.

Order	Insect Species	Cytosolic							Total	Ref.
		Delta	Epsilon	Omega	Sigma	Theta	Zeta	Unclassified		
Coleoptera	*Lasioderma serricorne*	1	0	0	1	1	0	0	3	[98]
	*Agrilus planipennis*	5	9	0	2	0	0	0	16	[98]
	*Anoplophora glabripennis*	10	10	2	4	2	0	0	28	[99]
	*Rhaphuma horsfieldi*	5	8	3	2	1	1	0	20	[98]
	*Xylotrechus quadripes*	5	7	2	2	1	1	0	18	[98]
	*Diabrotica virgifera*	3	11	1	0	2	0	0	17	[98]
	*Leptinotarsa Decemlineata*	6	11	7	6	2	1	0	33	[100]
	*Phyllotreta striolata*	5	6	2	6	1	1	2	23	[101]
	*Dendroctonus armandi*	0	4	1	2	1	0	0	8	[102]
	*Dendroctonus ponderosae*	6	12	2	5	2	1	0	28	[103]
	*Lissorhoptrus oryzophilus*	3	7	2	8	1	1	2	24	[98]
	*Sitophilus oryzae*	2	12	3	6	2	1	0	26	[104]
	*Aethina tumida*	3	19	1	7	1	5	7	43	[105]
	*Oryctes borbonicus*	4	5	3	15	3	1	0	31	[106]
	*Onthophagus taurus*	4	7	3	1	4	0	0	19	[98]
	*Nicrophorus vespilloides*	8	6	0	1	3	0	0	18	[98]
	*Asbolus verrucosus*	3	14	2	2	1	0	0	22	[98]
	*Tribolium castaneum*	3	19	3	7	1	1	2	36	[79]
	*Tenebrio molitor*	2	13	1	5	1	1	2	25	[107]
Diptera	*Chironomus riparius*	3	1	1	4	1	1	2	13	[108]
	*Aedes aegypti*	8	8	1	1	4	1	3	26	[109]
	*Anopheles gambiae*	17	8	1	1	2	1	2	32	[79]
	*Culex quinquefasciatus*	14	10	1	1	6	0	3	35	[110,111]
	*Drosophila melanogaster*	11	14	4	1	4	2	1	37	[79]
	*Bactrocera dorsalis*	9	5	3	1	3	3	1	25	[112]
	*Ceratitis capitata*	7	14	1	1	3	2	1	29	[113]
Hemiptera	*Bemisia tabaci*	14	0	1	6	0	2	0	23	[114]
	*Orius laevigatus*	1	0	2	16	1	1	0	21	[115]
	*Acyrthosiphon pisum*	16	1	2	6	2	0	3	30	[79]
	*Myzus persicae*	8	0	0	8	2	0	0	18	[116]
	*Laodelphax striatellus*	1	1	1	3	1	1	0	8	[117]
	*Nilaparvata lugens*	2	1	1	3	1	1	0	9	[118,119]
	*Supraphorura furcifera*	2	1	1	1	1	1	0	7	[119]
	*Rhodnius prolixus*	1	0	1	7	4	1	0	14	[120]
	*Diaphorina citri*	2	2	0	3	0	0	1	8	[114]
Hymeno-ptera	*Apis mellifera*	2	0	2	4	1	1	1	11	[79]
	*Bombus impatiens*	5	0	2	4	1	1	0	13	[121]
	*Bombus terrestris*	5	0	2	4	1	1	0	13	[121]
	*Meteorus pulchricornis*	4	0	3	7	0	1	0	15	[122]
	*Nasonia vitripennis*	5	0	2	8	3	1	0	19	[123]
	*Pteromalus puparum*	5	0	2	8	3	1	0	19	[124]
Lepidoptera	*Bombyx mori*	4	8	4	2	1	2	2	23	[125]
	*Cnaphalocrocis medinalis*	4	9	3	5	0	2	2	25	[107]
	*Heortia vitessoides*	3	2	3	3	1	2	2	16	[126]
	*Spodoptera litura*	5	21	3	7	1	2	3	42	[127]
	*Danaus plexippus*	3	6	3	5	1	3	2	23	[98]
	*Pieris rapae*	3	3	4	4	1	2	0	17	[128]
	*Plutella xylostella*	5	5	5	2	1	2	2	22	[129]
	*Manduca sexta*	6	9	4	2	1	2	1	25	[98]
	*Cydia pomonella*	4	3	2	1	1	1	1	13	[130]
Orthoptera	*Locusta migratoria*	10	0	3	12	2	1	0	28	[131]
Phthir-aptera	*Pediculus humanus*	4	0	1	4	1	1	0	11	[132]
Psocoptera	*Liposcelis entomophila*	17	0	1	13	3	1	0	35	[133]

**Table 3 biomolecules-13-00322-t003:** Identification of GSTs in the chemosensory organs of various insect species.

Order	Insect Species	Location	GST Classes	Ref.
Coleoptera	*Agrilus planipennis*	Antennae	Delta	[142]
	*Dendroctonus valens*	Antennae	Not indicated	[143]
	*Phyllotreta striolata*	Antennae	Delta and Epsilon	[101]
Diptera	*Aedes albopictus*	Antennae/maxillary palps	Not indicated	[151]
	*Drosophila melanogaster*	Antennae/maxillary palps/labellum	Delta, Epsilon, Omega, Sigma, Theta, and Zeta	[5,17,134]
Hymenoptera	*Apis melifera*	Antennae	Delta	[144]
Ixodida	*Dermacentor variabilis*	Haller’s organ	Epsilon and Mu	[145]
Lepidoptera	*Bombyx Mori*	Antennae	Delta	[137]
	*Chilo suppressalis*	Antennae	Delta, Epsilon, Omega, Sigma, Theta, and Zeta	[136]
	*Cydia pomonella*	Antennae	Delta, Epsilon, Omega, Sigma, Theta, and Zeta	[130]
	*Epiphyas postvittana*	Antennae	Delta, Epsilon, Omega, Sigma, and Theta	[152]
	*Grapholita molesta*	Antennae	Delta	[148]
	*Helicoverpa armigera*	Antenna	Delta	[139]
	*Heortia vitessoides*	Antennae	Delta and Epsilon	[126]
	*Manduca sexta*	Antennae	Delta	[140]
	*Papilio xuthus*	Antennae, labella, and tarsi	Delta	[153]
	*Plodia interpunctella*	Antennae	Delta, Epsilon, Omega, Sigma, Theta, and Zeta	[154]
	*Spodoptera littoralis*	Antennae	Delta, Epsilon, Omega, Sigma, Theta, and Zeta	[141,155]

## Data Availability

All data are available in the manuscript.

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
