# Peer review of "Role of Insect and Mammal Glutathione Transferases in Chemoperception"

_biomolecules, 2023, doi:10.3390/biom13020322_

Round 1
Reviewer 1 Report
Present review article aimed to provide knowledge supporting the involvement of GSTs in chemoperception, describing their localization in these systems as well as their enzymatic capacity toward odorants, sapid molecules and pheromones in insects and mammals. Their different roles in chemosensory organs will be discussed in view of the evolutionary advantage of the coupling of the detoxification system and chemosensory system through GSTs. In generally, the manuscript was well prepared. I suggest accept it after minor revisions as follow:
1. I suggest author also add one table to describe GSTs in mammalian chemoperception.
2. I suggest author add one part to describe the common GSTs both in mammalian chemoperception and insects; list the GSTs specific in mammalian chemoperception and in insect, and discuss the possible functions of GSTs specific in mammalian chemoperception or in insects.
3. The writing English should be improved by native English speaker. I strongly suggest to check the manuscript carefully, especially please check the grammar and the completeness of the sentences once again. And please check the tense of the sentences. There should be consistency throughout the manuscript using past tense.
Author Response
- I suggest author also add one table to describe GSTs in mammalian chemoperception.
As proposed we added a new table including the localisation of GST members in different mammal’s species.
- I suggest author add one part to describe the common GSTs both in mammalian chemoperception and insects; list the GSTs specific in mammalian chemoperception and in insect, and discuss the possible functions of GSTs specific in mammalian chemoperception or in insects.
Most of the GST classes are different between mammals and insects. Moreover, the most studied GST members in insects are the specific Delta and Epsilon class and for mammals, the specific Alpha, Mu and Pi classes. Then the few data concerning common GST members as the Zeta are too weak to discuss it in an evolutive context.
- The writing English should be improved by native English speaker. I strongly suggest to check the manuscript carefully, especially please check the grammar and the completeness of the sentences once again. And please check the tense of the sentences. There should be consistency throughout the manuscript using past tense.
The paper was corrected by American journal expert before submission. Moreover, English was improved in this new version.
Reviewer 2 Report
In abstract section, different types of letters are observed.
Lines 46 to 49 and lines 54 to 56 have repeated content.
In line 141 CO2... 2 is subindex
Please, to check that the scientific names are in italics (examples, lines 70, 86) and in vivo line 212, trans in line 371 and Z in name of alcohol in line 376.
Dipteran and lepidopteran instead of dipter and ledidopter. line385
In table 1 please, to incorporate the complete scientific name, similar to table 2. column of Total identified GSTs does not correspond to the sum of these (in second page of the table).
Different types of letters are observed in table 2.
Author Response
In abstract section, different types of letters are observed.
Thanks, now all the document is homogeneous.
Lines 46 to 49 and lines 54 to 56 have repeated content.
Thanks, we changed the second sentence
In line 141 CO2... 2 is subindex
It was changed
Please, to check that the scientific names are in italics (examples, lines 70, 86) and in vivo line 212, trans in line 371 and Z in name of alcohol in line 376.
It was changed
Dipteran and lepidopteran instead of dipter and ledidopter. line385
It was changed
In table 1 please, to incorporate the complete scientific name, similar to table 2. column of Total identified GSTs does not correspond to the sum of these (in second page of the table).
It was changed
Different types of letters are observed in table 2.
The table was corrected to be homogeneous
Reviewer 3 Report
The manuscript by Schwartz et al. try to report on the role of glutathione transferases in chemoperception of insect and mammal. But the organization and description is poor. There are a lot of irrelevant content, for example the “Chemoperception in mammals”, “Chemoperception in insects”. The manuscript should focus on the connection between the glutathione transferases and chemoperception, but lack of appropriate and adequate references. Therefore, I don't think this paper has enough evidence to support the view on the function of GST in chemoperception.
Author Response
Thanks for your opinion. To understand the common roles of GST in mammals and insects as presented in the discussion we need first to describe the chemoperception then the GST roles in this function. More than 20 original papers, published by various research teams show a role of GSTs in chemoperception. This review did not bring new evidences, however the GST role in chemoperception was propose in both insect and mammals, despite the fact that their chemosensory systems appeared independently. We hope that the comparison brought here will help the community to provide new experiments to convince you in the future.
Round 2
Reviewer 2 Report
The authors have addressed all of the comments from my review of the initial submission.
Author Response
Thanks a lot for your last advices. We paid a MDPI english editing on this last version.
Best regards
Fabrice Neiers